# Unusual Mathematical Approaches Untangle Nervous Dynamics

**DOI:** 10.3390/biomedicines10102581

**Published:** 2022-10-14

**Authors:** Arturo Tozzi, Lucio Mariniello

**Affiliations:** 1Center for Nonlinear Science, University of North Texas, Denton, TX 76203-5017, USA; 2Department of Pediatrics, University Federico II, 80131 Naples, Italy

**Keywords:** monocular cue, microcolumn, infinity topoi, globular set, embryonal neurulation, neurodata, geometry, topology, group theory, category theory

## Abstract

The massive amount of available neurodata suggests the existence of a mathematical backbone underlying neuronal oscillatory activities. For example, geometric constraints are powerful enough to define cellular distribution and drive the embryonal development of the central nervous system. We aim to elucidate whether underrated notions from geometry, topology, group theory and category theory can assess neuronal issues and provide experimentally testable hypotheses. The Monge’s theorem might contribute to our visual ability of depth perception and the brain connectome can be tackled in terms of tunnelling nanotubes. The multisynaptic ascending fibers connecting the peripheral receptors to the neocortical areas can be assessed in terms of knot theory/braid groups. Presheaves from category theory permit the tackling of nervous phase spaces in terms of the theory of infinity categories, highlighting an approach based on equivalence rather than equality. Further, the physical concepts of soft-matter polymers and nematic colloids might shed new light on neurulation in mammalian embryos. Hidden, unexpected multidisciplinary relationships can be found when mathematics copes with neural phenomena, leading to novel answers for everlasting neuroscientific questions. For instance, our framework leads to the conjecture that the development of the nervous system might be correlated with the occurrence of local thermal changes in embryo–fetal tissues.

## 1. Introduction

Mathematics displays the curious operational feature to be able to describe and predict biophysical issues, despite our scarce knowledge of the relationships between abstract models and descriptive experimental results [1,2,3]. Mathematics works in the real world, even if we do not know why [4,5,6]. It has also been used as a unifying language/framework for the description of neuroscientific issues. Different mathematical fields have been used to cope with neural matters and to tackle the cognitive functions of the brain. In particular, computational neuroscience has provided an effort to go through the intricate matters of the neural activity at different coarse-grained levels [7,8,9,10]. Our approach in this paper will be slightly different. Our objective is not to discuss mathematics applied to neurodata. We will not consider the areas of calculus and analysis, i.e., the mathematical branches encompassing multivariable calculus, functional analysis and numerical analysis that allow the computation of the ordinary/partial differential equations arising in many neuroscientific applications. We will not focus on the mathematics subtending the implementation of various neurotechniques such as EEG, fMRI, network reconstruction, etc. [11,12,13,14,15,16,17,18]. Rather, our main objective is to highlight unnoticed relationships linking mathematical abstract concepts to the neuronal activity which subtends the very function of the central nervous system.

We will proceed as follows. Every chapter will be devoted to a different area of neuroscience. Macro-, meso- and microneuroanatomy, nervous development and visual perception will be tackled in purely mathematical terms. In particular, approaches from manifold mathematical branches, such as geometry, topology, group theory and category theory will be used to tackle these issues. We will describe the neurobiological counterparts of braid groups, elliptic curves, micronetworks, liquid crystals, Monge’s theorem, cohomology groups and shaves. We will provide a survey of mathematical methodologies and their neuroscientific counterparts that might contribute to improve our knowledge of the central nervous system. Furthermore, we will suggest next-to-come, feasible mathematical developments which may prove potentially helpful in the experimental assessment of brain dynamics.

## 2. Mathematics and the Anatomy on the Central Nervous System

In this chapter, we will describe how mathematical concepts might be useful to quantitatively assess anatomical structures of the central and peripheral nervous systems. We will provide three examples and different coarse-grained anatomic scales. First, we will show how knots and braid groups provide a scarcely explored approach to elucidate the arrangement of nervous fibers at the mesoscopic scales of the ascending pathways. Second, we will show that the mathematical concept of elliptic curves might be used to evaluate interhemispheric connections at mesoscopic scales. Third, we will show how the successful mathematical approaches based on neural networks might be unexpectedly extended to the microscopic scale of neuronal membranes.

### 2.1. Macroscopic Scale: Braid Groups, Nerve Fibers and Somatotopic Maps

Here the focus is focused on the anatomical intertwining of fibers in both the central and peripheral nervous systems. The description of intermingling peripheral nerves and mingling grey/white matter can be appraised in terms of knot theory and braid groups [19]. Knots can be turned one into another via three-dimensional space deformations, giving rise to assemblies with associative and commutative properties [20]. A braid is a collection of strands between two parallel planes [21]. Braids are termed isotopic when, keeping their endpoints fixed, they can be twisted into each other without cutting the strands. The operations of composition allow braids to be joined to achieve new ones. Given a set of braids with a fixed number of strands, its group structure is provided by generators and fusion rules such as, e.g., associativity, crossing, braiding/unbraiding, intertwining, the composition and sequence of elementary braids, and so on. In algebraic topology, a well-established link does exist between braids and knots. Is it always possible to transform a knot into a closed braid? For example, the Alexander theorem states that every knot or link in three-dimensional Euclidean space is the closure of a braid [22]. Nevertheless, the correspondence between knots and braids is not one-to-one because a single knot may have many braid representations. The Markov theorem provides the moves to relate closed braids representing the same knot type [23].

Biology suggests several examples of living structures that stand for feasible counterparts of mathematical knots and braids. For example, tissue morphogenesis is produced by coordinated regional changes in cell shape driven by localized contractions of actomyosin “braids” [24]. The existence of a wire-like flow of electrons and ions along cytoskeletal elements conveying messages from the cell membrane to the nucleus has been demonstrated [25]. The three-dimensional space of neural connections is severely constrained by physical factors, such as the anatomical overlap of neuronal arbors and the available axonal space to make synapses. Indeed, specific neuron-type patterns of distance-dependent connectivity are correlated with peculiar overlaps between the dendritic and axonal ramifications, so that the diverse branching patterns of individual arbors of the same neuronal type is able to influence higher-order connections [26,27].

Incorporating the mathematical/topological perspective of knot theory in neuroscience appears particularly relevant for understanding peculiar brain functions such as, e.g., how the brain represents and processes environmental stimuli. The arrangement of nervous fibers might stand for an example of isotopic braids describing the evolution of multi-particle systems with two spatiotemporal extremities: a beginning and an end. Just as the knots are embeddings of closed lines in the three-dimensional space and cannot be reduced to simple circles by a continuous deformation [28], the nerves are structures that do not completely disentangle after being pulled from both ends. The nervous fibers connecting distinct structures of the peripheral and the central nervous systems are shaped as braids. For example, consider the sensory inputs from the external world detected by the peripheral nervous fibers: these inputs follow a multisynaptic, ascending path towards the more proximal areas of the central nervous system. See Figure 1A,B for further details. This means that, if we term “braid group” as the whole nervous tract between the peripheral receptors and cortical areas, we are allowed to term “knots” as the intermediate multisynaptic steps. Figure 1C tells us that the anatomical nervous structures studied by tractography can also be described in terms of braids.

The overwhelming complexity of the mammalian nervous system makes It extremely difficult to map nervous anatomical structures to mathematical manifolds equipped with knots and braids. Given the objective obstacle to find the proper group generators, why should we care to describe the nervous paths in the tricky terms of braid groups? What are the (methodological, philosophical, explanatory, medical) benefits? Our suggestion is twofold:(a)Simple changes in the location and arrangement of nerves could explain the activity of the central nervous system.(b)The external inputs follow specific nervous paths which are assessable in the mathematical terms of braid groups.

The theoretical occurrence of a link between the anatomical conformation of the nerves expressed in terms of knots/braids and brain activity would mean that different braiding rules might give rise to different nervous functions. Fully different perceptions such as olfactive, auditory, tactile and visive perceptions could be explained by diverse configurations of the braids connecting every external receptor to the corresponding sensitive cortex. In touch with the suggestion that sensitive pathway conformations might affect cue perception, Guillamón-Vivancos et al. [29] suggest that, despite the tactile information simultaneously activating the tactile and visual neural pathways during the embryonic stage, the pathways reorganize after birth to permit the separate processing of visual and tactile information.

It is well-established that neural pathways are topographically organized in maps that are highly preserved from the periphery to the cortex [30]. This could be explained by the occurrence of isotopic braids in nervous fibers: as the strands between the starting and the ending points (i.e., the nervous fibers between the peripheral receptor and the corresponding cortical projection) display the same intermingled conformation, they might carry the same message. This hypothesis does not require that the external message stands strictly in a 1:1 relationship with the corresponding cortical area. Due to the braid rules, the relationship between the two extremities of the braids could be either injective or surjective, bijective and so on. Therefore, in a mathematical braid framework, the arrangement of the cortical somatotopic maps could be correlated with the arrangement of the peripheral nervous fibers. The connections between the two extremities might give rise to reversible or irreversible topological knots, respectively, standing for labile functional activities and stable anatomical tracts. This means that at least a part of human brain diseases might depend on the anatomical configuration of peripheral nerve fibers. Instead of looking for alterations in the central nervous systems, it would be reasonable to look for neuropathological features in the braids and knots of the peripheral nervous fibers.

In sum, braid groups could be the key to explaining the perceptual variations between different sensitive cues such as sight and hearing. Indeed, different braids conformations might generate distinct computational processes in the corresponding primary sensory cortex.

**Figure 1 biomedicines-10-02581-f001:**
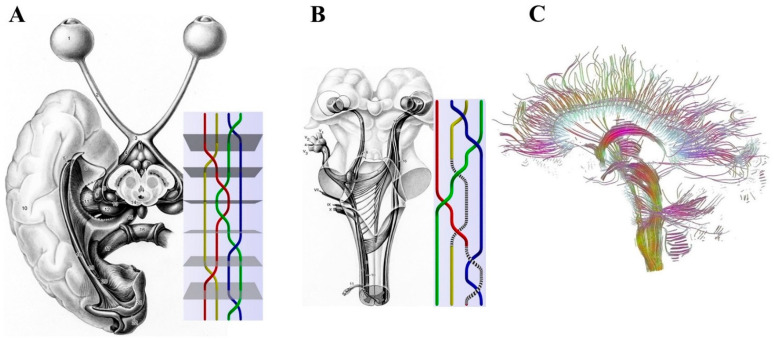
(**A**,**B**). Comparison of nervous connections in the central/peripheral nervous systems and mathematical examples of braid groups. The figures illustrate the retino–geniculo–cortical projection in a ventral view (**A**), the central connections of the trigeminal nerve in a sagittal view (**B**) and their hypothetical braid counterparts. Modified from: [31] For further details, see [32]. (**C)**. The anatomical nervous structures detectable by tractography can be described in terms of braids. Modified from: [33]—“Brain dataset courtesy of Gordon Kindlmann at the Scientific Computing and Imaging Institute, University of Utah, and Andrew Alexander, W. M. Keck Laboratory for Functional Brain Imaging and Behavior, University of Wisconsin-Madison”.

### 2.2. Mesoscopic Scale: Are There Elliptic Curves in the Brain?

Elliptic curves, generated through cubic equations, are characterized by two-dimensional paths devoid of either cusps or intersections. Elliptic curves are enclosed in two-dimensional finite algebraic fields. They can be defined in terms of points, integer numbers and rational numbers [34,35]. Many neurotechniques, e.g., EEG, fMRI, diffusion tensor imaging and diffusion tensor tractography, describe nervous structures (such as tracts, commissures, fasciculi, radiations) arranged as arcs that roughly resemble elliptic curves [36]. Starting from this remark, a mathematical account can be drawn which hypothesizes the occurrence of abstract structures underlying the anatomy and function of mesoscopic nervous pathways. The question is: what for? What do elliptic curves bring to the table when investigating nervous paths? Elliptic curves might stand for the abstract counterpart of the anatomical neural projections endowed in the finite field of the central nervous system. A brain including elliptic curves could be operationally partitioned in numbered areas containing either integer or rational numbers. This approach would allow us to treat these brain areas via the powerful weapons of algebraic geometry, complex analysis, number theory and representation theory. Being elliptic curve abelian [37], they must be equipped with intrinsic symmetries that are hidden at first sight. The occurrence of hidden symmetries would allow long-range, simultaneous activation of neurons located in distant brain areas. It is of note that half of the elliptic curves display a finite number of rational numbers, while the other half display an infinite number of rational numbers [38]. In operational terms, this means that half of the nervous patterns are continuous and half are discontinuous, i.e., arranged in discrete spatiotemporal steps. Last but not least, the solutions of the elliptic cubic curve are confined to a spatial region that is topologically equivalent to a torus. Therefore, as suggested by Tozzi et al. [39], the anatomical and functional nervous trajectories could be assessed in terms of trajectories occurring inside a torus.

### 2.3. Microscopic Scale: Towards a Transient Microconnectome Made of Tunneling Nanotubes?

Standard neuroscience textbooks tell us that neurons communicate via axons, dendrites and synapses, generating a sort of holistic structure that can be assessed via the mathematical weapons of network theory and dynamical systems theory. In dynamical systems, simple elements spontaneously aggregate into larger and more ordered structures, giving rise to different self-sustaining waves such as traveling waves, rotating waves and standing and reflected waves/spirals [40]. In nonequilibrium systems characterized by the self-organization of collective particles, input-driven local fluctuations lead to the emergence of larger-scale ordering [41,42]. This is the case not just of artificial devices such as shape-changing robotic active matter but also of biological assemblies [43]. For example, cellular self-organization promotes the follicle pattern in avian skin [44]. Self-organization generates processes such as crystallization which contributes to unusual self-assemblies of small and rigid organic molecules (such as proteins, viruses, nucleic acids, nervous structures), to colloids gathering in liquid crystals, to gold nanocrystal superlattices, etc. [45]. It has been suggested that the brain is a dynamical system at the edge of chaos characterized by random fluctuations [46,47,48,49,50]. Here the human connectome comes into play, i.e., the hierarchical anatomical and functional network of the cortical/subcortical structures that can be mathematically described by net theory. The connectome is characterized by random walks, preferential pathways for fast communication and winner-takes-all mechanisms [51,52,53,54].

Recently discovered microscopic entities could deeply modify the current paradigm of the neural connectome. Tunneling nanotubes (TNTs) are F-actin-based, transient tubular connections [55] that allow the active transfer of vesicles, organelles and small molecules between adjacent cells [56,57]. The occurrence of TNTs in primary neurons and astrocytes has been theoretically correlated with the short-range transmission of electrical signals [58,59,60]. Developing neurons form transient TNTs that both enhance electrical coupling with distant astrocytes and allow the transfer of polyglutamine aggregates between neuronal cells [61]. TNTs are not stable structures as their lengths vary with the distance between the connected cells [60], and their lifetime may range from a few minutes up to several hours [62,63]. The occurrence of TNTs with their peculiar transient features changes our notion of the human connectome made of stable nodes/edges and long-standing connections between brain areas [64]. Indeed, TNTs might provide a microscopic intercellular neural network with peculiar features, namely, the nodes are not stable and the edges appear, modify and vanish with passing time. As the extremities of TNTs link two cellular structures, the dogma of the cell as an individual unit might be questioned [57]. A microscopic narrative of connectome also provides an alternative explanation to the observation that some network branches are visited more than others, generating nervous dynamics which are slightly nonergodic.

## 3. Mathematics and the Embryonic Development of the Nervous System

Soft-matter polymers and embryonal neurulation. The development of multicellular living beings involves morphogenetic processes that shape embryo–fetal structures through the self-organized activity of pluripotent stem cells [65]. The orchestrated movement of cellular groups requires genetic as well as mechanical and molecular interactions between cells and their surrounding environment [65,66]. Migrating cells respond to various stimuli such as cortical tension, luminal pressure and size [67], local changes in tissue architecture [68] as well as topographical, adhesive and chemoattractant cues [69,70,71,72] and duplication of existing regions [73]. Changes in substrate stiffness trigger collective cell migration, suggesting that tissue mechanics combines with molecular effectors to coordinate morphogenesis [66]. We, focusing on the morphogenetic dynamical processes characterized by cooperative interaction and the collective migration of numerous cellular units [41], describe here an unnoticed biophysical process that might underlie the very structure of the systems of vertebrates. The embryo–fetal development of the central and peripheral nervous systems is of foremost importance as the growth of the neuronal tissue is correlated with the growth of a wide range of non-nervous structures [74].

Brain-derived signals are involved in embryogenetic regulation, providing long-range interactions among separate structures. During embryogenesis, unexpected interactions have been described between the nervous system and numerous craniofacial and trunk skeletal elements [75,76]. For instance, multipotent Schwann cell precursors detach from their nerve fiber commitment to become mesenchymal, chondroprogenitor and osteoprogenitor cells [77]. In addition, a link has been found between two apparently unrelated processes, gastrulation and neural crest migration, via changes in tissue mechanics [66]. In the sequel, we aim to portray embryonal neurulation in terms of condensed colloidal self-matter and its liquid/crystalline phase.

Cellular structures could be viewed as building blocks characterized by liquid/crystalline phases of condensed soft matter, in which order and fluidity coexist [78]. Liquid–liquid phase separation allows intracellular organization within distinct compartments of bacteria and eukaryotes [79,80]. The phase transitions drive proteins to aggregate into cytoplasmatic and nuclear-condensed fluid bodies, generating nonuniform localization patterns and subcellular compartmentalization. Biomolecular condensates include membrane protein clusters, cytoplasmic P granules [81], histone locus body, heterochromatin domains [82], protoplasmic gelation [83], amyloid-like assemblies [84] and intrinsically disordered mixed-charge domains. Liquid-phase condensates can be viewed as reaction centers where some components become enriched for processing or storage within cells. For example, Garcia Quiroz et al. [85] found that the keratinocytes of the stratified squamous epithelium undergo a vinegar-in-oil type of liquid–liquid phase separation, crowding the cytoplasm with increasingly viscous protein droplets able to drive squamous formation.

Compounds made of “liquid crystals” display properties between conventional liquids and solid crystals that can be experimentally studied in vitro [86,87,88,89,90]. Mundoor et al. [91] produced building blocks of a molecular–colloidal liquid crystal made of micrometer-long inorganic silica-coated disks, dispersed in a crystalline fluid composed of molecular rods. Field-induced motion caused by the magnetic fields elicited colloidal interactions between the disks in the nematic hosts and generated various symmetric conformations with different tangential surface orientations (Figure 2A). Within a range of temperature and concentration, the freely diffusing rods arranged themselves orthogonally to the solvent molecules, producing a biaxial liquid crystal. Mundoor et al. [92] demonstrated that the dispersion of the isotropic-charged colloidal disks in the nematic host composed of molecular rods produces isotropic, nematic and smectic columnar organizations (Figure 2A). While regular polymeric materials respond in a linear fashion to external stimuli such as high temperature, liquid crystal polymers display nonlinear and much faster macroscopic changes [93]. It is feasible to realize low symmetry condensed matter phases in systems with building blocks of dissimilar shapes and sizes. During the embryo–fetal differentiation of the central nervous system, the bodily architecture recalls isotropic, nematic and smectic columnar arrangements [94]. See Figure 2B for further details. It is noteworthy that biaxial nematics can be produced either through long inorganic nanorods and short organic molecules, or board-like molecules or component mixtures, paving the way to future approaches focused on organic structures. The experimental clues point towards a relationship between the biological elements (in our case, the nervous elements) and the phase changes, e.g., the assembly of the developing synaptic active zone requires the liquid phase of the scaffold molecules [95].

Summarizing, we suggest investigating the relationship between the processes of liquid crystals arrangements and the formation of embryonic nervous structures. To further scrutinize this hypothetical correlation, we provide a testable hypothesis. The collective phenomena of colloidal interactions between disks in the nematic hosts described by Mundoor et al. [91,92] are thermotropic, i.e., they are temperature-driven and temperature-dependent. Increases in temperature in the nematic colloidal fluids lead to unusual transitions towards more ordered states. When the temperature is lowered, symmetry-breaking phase transitions lead at first to the transition from isotropic liquid to nematic phases and then to a liquid–crystal smectic phase [78]. In touch with classical polymer physics, Kießling et al. [96] observed a systematic deformation of the viscous cellular matter upon temperature changes. This framework enables us to draw an intriguing suggestion that correlates the differences in cellular temperature with the developmental outcomes of the nervous system. Within biological cells, transient temperature spikes and short-distance heat fluxes have been detected [97]. Such nonstationary local fluctuations in cellular temperature are also worth exploring in the nervous system. Indeed, differences exist between the temperatures of the cell body and neurites [98]. Further, the neocortex displays thermal gradients at various spatiotemporal coarse graining [49]. The membrane temperature can modify the neuronal activity via adjustments in the opening and closing rates of ion channels [99]. Local changes in temperature can modulate presynaptic and postsynaptic events, sensitive stimuli and memory encoding [100,101]. As fluctuations in thermic flows are correlated with variations in thermodynamic/information entropies, a correlation might also be hypothesized between temperature changes and the message content [102,103].

Summarizing, the development of the central and peripheral nervous systems could be correlated with the local thermal changes occurring in embryo–fetal tissues. The fact that the embryonal neurulation occurs at different temperatures in different animals does not invalidate our theoretical account as every species-specific embryogenic temperature might lead to anatomical modifications in the adult nervous system.

## 4. Mathematics and Visual Perception

In this chapter, we will describe how mathematical approaches might be helpful to assess the features of visual perception. The research area of visual perception is huge, including, e.g., the modelling and control of visual perception, visual perception learning, static and motion-based visual illusions and visual–perception models related to the mathematical aspects of dynamic visual cryptography [104,105,106,107,108,109,110]. Here we will show how the Monge’s theorem might provide a novel operational approach to the dazzling phenomenon of depth perception (DP). DP, crucial for everyday action, is the visual ability of humans and other sighted animals to perceive both the distance of an object and its three-dimensional solid structure [111]. Many anatomical structures and sources of information contribute to produce DP in different animals. Among the most important mechanisms subtending DP, binocular and monocular perception are worth a mention [112].

Binocular depth perception relies on two main mechanisms, i.e., stereopsis and retinal binocular disparity [113,114]. Stereopsis depends on the fact that many animals adjust their two eyes to place the image of an object on the fovea, where acuity is the highest. In turn, retinal binocular disparity suggests that, as our eyes are separated, the images of the object fall at different locations on the left and right retinas [115]. The visual system melts these two images to quantify the depth of the location of the object. In turn, monocular depth perception provides depth information when viewing a scene with one eye [116]. Many sources of information from the image and its surrounding environment produce monocular DP, including, e.g., relative size (distant objects subtend smaller visual angles than near objects), texture gradient, occlusion, contrast differences (e.g., texture gradient, illumination and shading), motion parallax, etc. [117]. To make an example, the first studies on DP involved aircraft pilots during take-off and landing, showing that texture analysis and motion detection are the most relevant features for the perception of solid structures [118]. For depth perception at close range, stereopsis is the most precise mechanism but is available only to a few vertebrates. In turn, motion parallax is used by many species, including vertebrates as well as invertebrates. Compared to other animals, humans display the best performance regarding depth resolution [119].

In the last decades, it has been demonstrated that shape detection also depends on higher cognitive activities such as previous experience/memory, context-dependent size interpretation and whether the observer is “expecting” to see the shape. Accumulating evidence suggests that the main three computational goals (determining surface depth order, gauging depth intervals and representing three-dimensional surface geometry and object shape) are provided by different hierarchical stages of cortical processing [120]. The integration of right and left eye information generates cortical three-dimensional representations of the visual environment [121]. For instance, in mammals such as mice that use stereoscopic cues to guide their behavior, neurons in the primary visual cortex V1 are sensitive to binocular disparity. In sum, the complex response of DP is built up from multiple environmental inputs and different intermingling distal and proximal nervous processes.

Here we suggest another novel monocular mechanism that might contribute to DP. Our approach is based on a theorem from geometry, the Monge’s theorem (MT). MT states the following: given three nonoverlapping circles of distinct radii in a two-dimensional Euclidean plane, the intersection points of each of the three pairs of external tangent lines lie on a single line [122]. See Figure 3A for a pictorial rendering.

MT implies that the projections arising from three separated objects (i.e., two- or three-dimensional circles) give rise to a line that is external to the three objects. MT has been operationally used to calculate the correct camera position for the wearing examination of the cutting edge of the hob, by calculating the inner points of the Monge cuboid and their parallel shifting (defined by the bijective Monge projections) bordered by a surface [123]. Further, MT is still valid even if the plane is equipped with non-Euclidean metrics [124].

**Figure 3 biomedicines-10-02581-f003:**
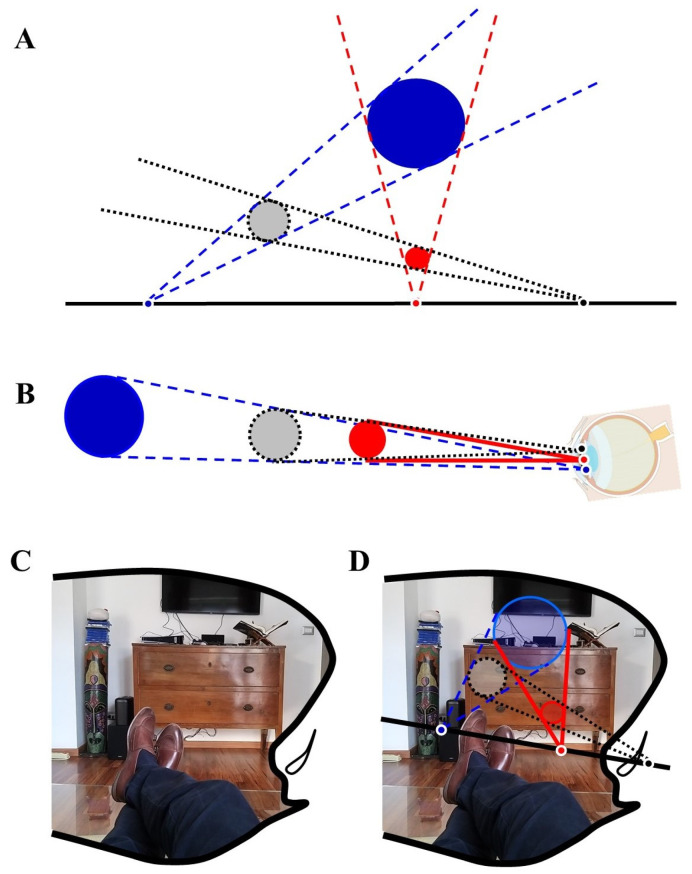
Monge’s theorem (MT) for the evaluation of depth perception. (**A**) Pictorial rendering of MT. (**B**) MT can be used to investigate the physiological mechanism of depth perception. The line on the eye lens corresponds to the line joining the projections from three objects embedded in the environment. (**C**) A subject lying upon his sofa with his right eye closed. “In a frame formed by the ridge of my eyebrow, by my nose and my moustache, appears a part of my body, so far as visible, with its environment” [125]. (**D**) An example of the monocular depth perception of MT drawn from (**C**). Three objects with different distances from the eye are projected to a black line on the eye lens, according to the MT rules.

We suggest locating the line external to the three objects in the eye lens (Figure 3B). This line is produced by three sensed objects located in the environment. Due to the MT, the projections of the three objects on the eye lens are slightly far apart from one another. The line is subsequently projected from the eye lens to the retina, giving rise to retinal objects that are slightly far apart from each other. This simple, monocular mechanism might stand for a mathematical device that promotes our three-dimensional perception of the world (Figure 3C,D).

## 5. Discussion

Our theoretical account emphasizes how mathematical rules might shape the structure and activity of biological entities. The theoretical implications of our account suggest that simple changes in the arrangement of the anatomical and functional nervous structures might elucidate (at least partially) the activity of the central nervous system. In touch with D’Arcy Thompson [65], it could be stated that the arrangement and the pattern contribute to the biological function. Geometric constraints are powerful factors which are able to define shape, size and cell distributions, drive crucial biological phenomena and give rise to deterministic patterns that are predictable and reproducible. For example, methods to develop the spatial and temporal control of stem cell-derived epithelial organoids have been described, thereby rendering a stochastic process more deterministic [126]. A mathematical-framed approach to scientific matters provides a metatheoretic starting point that might be termed “testable rationalism”: sharp experimental previsions arising from top–down, deductive mathematical approaches. Mathematical weapons such as group structure and generator operations point towards a new approach to long-standing questions concerning human sensation and perception. This leads to the hint that the very intermingling of nervous structures might contribute to brain functions. As the dynamical processes of living systems display cooperative interaction of many units, we are allowed to portray the development of the central/peripheral nervous systems in terms of assemblies of building blocks dictated by mathematical constraints.

We want to give a further theoretical suggestion, this time drawn from category theory and group theory [127,128,129,130]. Once a wheat sheaf is sealed and tied up, the packed-down straws display the same orientation. This trivial observation brings us into the realm of presheaves/globular sets that allows a simple assessment of diverging and superimposing functions. A mathematically well-founded assessment of elusive nervous activities in terms of presheaves as well as the hierarchical information transmission inside globular sets provide fresh insights on different neural issues. Presheaves also permit the tackling of the nervous phase spaces in terms of the theory of infinity categories, i.e., an approach founded on equivalence, rather than equality [131]. The key to unlocking the activity of the brain does not lie in single neurons, or single neuronal assemblies or single neural modules, rather in its abstract functional phase spaces where equivalence and not identity holds. Here we will explain this time in technical terms, a novel theoretical approach to nervous paths based on infinity topoi. This permits us to introduce Lurie’s theory of infinity topoi [132] to describe and quantify the paths taking place inside the brain phase spaces. As sheaf-theoretic methods are powerful and flexible enough to allow generalizations of neural paths, Lurie’s theory enjoys formal properties that suggest novel functional phase spaces where brain activities might take place.

Every category of sheaves of abelian groups contains derived functors called cohomology groups [133] that can be defined as follows:H^n^ (X,G)
where X is a topological space, and G an abelian group [131].

This means that the cohomology group H^n^ (X,G) can be defined in terms of sheaf cohomology:H^n^_sheaf_ (X,G)
where G stands for the constant sheaf on X. Remember that a sheaf is a presheaf that satisfies the gluing axiom.

Lurie [131] generalized this approach to non-abelian cohomology H^1^ (X,G), also taking into account the fact that the coefficient system G is not constant and may display different values. This is the case for the brain phase spaces. Specifying G is equivalent to specifying the Eilenberg–MacLane space K(G,1), together with a base point. This observation suggests that the proper coefficients G for non-abelian cohomology H^1^ (X,G) are not groups but rather homotopy types, i.e., purely combinatorial entities such as simplicial sets. Sheaves of homotopy types on X can be used as coefficients, achieving a theory of infinite stacks (in groupoids) on X. Stacks, that are required to satisfy a descent condition only for covering, satisfy the Whitehead theorem: a pointed stack (E,η) can be achieved, for which π_i_(E,η) is a trivial sheaf for all i > 0, such that E is not contractible. If K is a simplicial set, then the cohomology of X with coefficients K can be defined as:H_jj_ (X,K) = π_0_ (F(X))
where H_jj_ is the Joyal–Jardine homotopy theory of simplicial presheaves on X, and F is a fibrant replacement for the constant simplicial presheaf with value K on X [131].

In brain terms, if X stands for the manifold where nervous oscillations take place, and G stands for an operation performed by the brain (say, the oscillatory activity), the following conclusions can be drawn together with Lurie [131]:(1)If X is paracompact, H (X, K) is the set of homotopy classes from X into K.(2)If X is paracompact space of finite covering dimension, then Lurie’s theory of stacks is equivalent to the Joyal–Jardine homotopy theory.

The definition of a sheaf depends just on the open sets of a topological space, rather than the individual points; this means that open sets could be replaced by other objects. The stalk F_x_ of a sheaf F captures the properties of a sheaf “around” a point x ∈ X, generalizing the germs of functions.

In neuroscientific words, we can state the following: even if one looks at smaller and smaller neighborhoods when single neuronal assemblies or tiny brain areas are investigated, no single neighborhood is small enough such that some limit can be taken into account [134]. In terms of nervous issues, this means that a nervous object of investigation, e.g., either the single neurons, or the single brain modules or an assembly of neural waves with the same frequency, stand in relation to each other in many ways. Single neural structures do not count anymore, rather the whole brain activity counts. Summarizing, an infinity topoi approach to neural dynamics suggests that nervous activities can be described in terms of sheaves and presheaves leading us into the realm of the ∞-topos, i.e., a ∞-category of ∞-stakes on a topological space that is correlated to ordinary topos. We are faced with two entirely different approaches to brain dynamics:(1)The customary concept of equality suggests the occurrence of a strict relationship between two entities (say, two neurons or two neural waves on the brain surface).(2)Lurie’s concept of equivalence of ∞-topos suggests that two entities (say, two neurons or two neural waves on the brain surface) stand in relation to each other in many ways.

In this latter account, the relationships between the entities can be studied in terms of the different forms correlated with homotopy paths, explaining why apparently matching cortical oscillations give rise to highly different interindividual responses.

## 6. Conclusions

We went through unusual mathematical approaches to assess several aspects of the nervous activity. The translational implications of our account are manifold. For instance, the Monge’s theorem suggests a new methodological approach to cope with our visual ability of depth perception. Further studies are required to confirm whether the human brain might use this theorem to achieve visual depth through monocular vision. This would lead to a better comprehension of the peripheral visual mechanisms and would suggest that the peripheral receptors perform advanced computations previously believed to be of exclusive pertinence to the central nervous structures. Our suggestion that the brain connectome can be tackled in terms of tunnelling nanotubes sheds new light on the studies concerning the human connectome. We propose that the human connectome can be tackled not just at the macrolevel of the connection of brain areas or at the mesolevel of neurons linked by axons/dendrites but also at the microlevel of intermembrane connexions among neuronal calls. This suggests a highly reductionistic approach where factors of the size of angstrom may dictate macroscopic behavior and pathological outcomes. The fact that the multisynaptic ascending fibers connecting the peripheral receptors to the neocortical areas can be assessed in terms of knot theory/braid groups suggests a novel approach to predict fiber regeneration during abnormal human nervous development and after trauma. Further, the physical concepts of soft-matter polymers and nematic colloids applied to neurulation in mammalian embryos lead to the conjecture that the development of the nervous system might be correlated with local thermal changes in embryo–fetal tissues. It suggests that the artificial embryos created for experimental and translational purposes can be manipulated through the relatively simple local modification of temperature.

Showing that natural three-dimensional natural rock fragments reproduce the Plato’s cube, Domokos et al. [135] found that distinct fragment patterns tend towards ubiquitous, standard icosahedral and octahedral shapes which can be formulated in terms of Archimedean lattices [136]. The spontaneous occurrence of distinctive self-assembled artificial structures could be compared with the design of natural organic crystalline materials, such as the nervous structures. In particular, both artificial and natural quasicrystal structures can generate fullerenic-like self-assemblies grounded on simple, overarching geometric rules [45,137,138,139,140,141,142,143,144,145]. The same fullerenic structures have been recently proposed to explain the features of cortical microcolumns that can be flattened to form fullerene-like, two-dimensional lattices [146,147]. As a final remark, we would like to emphasize that our experimental predictions can be demonstrated via current technologies. Concerning, for example, the theoretical correlation between the embryonal mechanisms of nervous growth and liquid crystals, the recent availability of artificial embryonic structures from the aggregates of mammalian stem cells [148,149,150,151,152] provides an exciting possibility to study in vitro the developmental processes.

In sum, we suggest that under-rated geometrical and topological concepts may not just shed new light of the dynamics of the central and peripheral nervous systems but may also provide hints towards feasible translational applications.

## Figures and Tables

**Figure 2 biomedicines-10-02581-f002:**
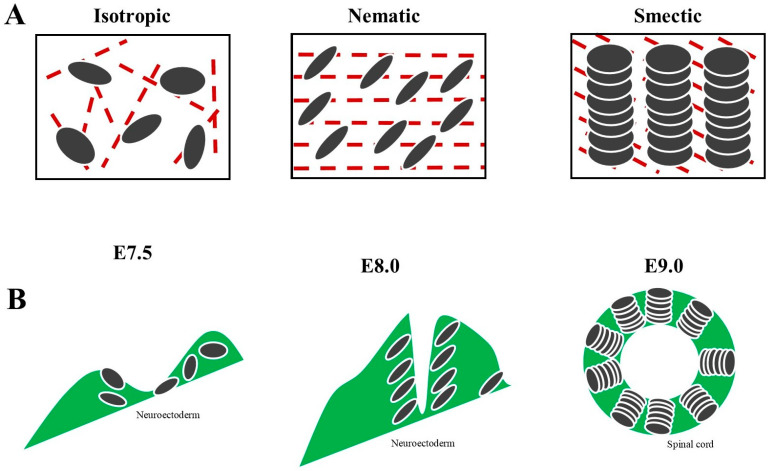
Comparison of two seemingly uncorrelated phenomena, i.e., liquid crystal phases and embryonic development of the nervous system. (**A**) Mixtures of molecular/colloidal rods and disks of fluid-condensed matter give rise to temperature-dependent columnar chains displaying different uniaxial symmetries. Depending on the structural arrangement, we achieve isotropic, nematic and smectic liquid crystals. For further details, see [91]. (**B**) Schematic transverse sections of neurulation in the mouse embryo at different stages of development. While the primitive confined neuroectoderm at E.75 recalls isotropic liquid crystals (**left picture**), the converging neural folds at E8.0 remind the arrangement of nematic liquid crystals (**middle picture**). In turn, the spinal cord at E.9.0 evokes the typical arrangement of smectic liquid crystals (**right picture**). For further details, see [94].

## Data Availability

All data and materials generated or analyzed during this study are included in the manuscript. The authors had full access to all the data in the study and take responsibility for the integrity of the data and the accuracy of the data analysis.

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
