# Peer review of "Unusual Mathematical Approaches Untangle Nervous Dynamics"

_biomedicines, 2022, doi:10.3390/biomedicines10102581_

Round 1

Reviewer 1 Report

This manuscript proposes various concepts linking basic mathematical theories to some interesting topics (mostly, anatomical issues) in the nervous system.  The writing is good, although the topic is hypothetical. The mathematics includes geometry, topology, knot theory, braid groups, category theory, group theory, and even algebra and number theory. It also contains some cell biological, developmental biological, and physical phenomena.  

This manuscript has no experimental data, but the authors tried to formulate testable ideas. Those ideas were not experimentally proven in this manuscript, or were not subjected to meta-analysis based on previous data. They could be right or wrong. Some people may not feel that they are scientifically sound.  However, I enjoyed the authors’ ideas because many of them are inspiring. Most important, they are scientific.  Thus, I would hesitate to raise any concerns about each hypothesis that the authors proposed here. I have two major suggestions to help the readers understand the authors' proposals. 

First, I often felt that some ideas are presented without enough neuroscience background.  The authors often introduced theories, and then brought up real-world biology. Thus, it sounds strange at some places.  In general, I would suggest placing more standard neuroscience and biology (noticed ideas) before the authors introduce their “unnoticed” ideas. In this way, the authors’ ideas will be readily recognized as “unnoticed” ideas. 

Second, the authors tend to consider only humans.  However, other species also have similar structures.  I would suggest adding this aspect so that the authors' ideas will be more influential.  

Minor points

Line 46-51  Here, I wonder why the authors organized topics in this order.  Is there any scientific reason for this? Otherwise, this manuscript looks like a random collection of essays in which a series of interesting ideas are randomly arranged. 

Line 52 onwards, Geometry chapter

Monge’s theorem is interesting. There are many explanations about binocular and monocular visions (eg. saccade).  Most important, the authors consider visions of only human.  How about other species (eg, pigeon, owl)?  

Line 95 onwards, Tunneling nanotubes chapter

This chapter focuses on a cell biological structure.  Thus, this part is different from other parts where the authors advocate mathematical theories. 

Standard neuroscience textbooks usually tell that neuron communicate via axons, dendrites, and synapses.  However, the authors do not provide this standard textbook view.  Many cell biologists would agree that we cannot dismiss tunneling nanotubes.  However, TNT should be presented together with the standard neuroscience view. 

Line 110 nervus -> nervous

Line 146 onwards, Topology chapter (especially, line 186-)

The topological or topographical organization of neuronal connections is commonplace.  For example, in lower vertebrates, it is clear that topology plays important roles in visual perception. Here, I feel that the authors’ argument is not well balanced (line 175-).  

Line 184, Several Authors -> Several authors

Line 194 chapter knot theory and braid groups 

The idea sounds interesting.  However, it is not so clear why knot theory and braid groups help understandneuroscience as the authors brought up (Line 246-).  The authors tried to explain their benefits. However, the benefits are not convincingly addressed because it looks the authors just presented a list of phenomena and theory. 

Line 281- category theory and group theory

It is not clear that "Soft matter polymers and embryonal neurulation" is in "category theory and group theory". I like the testable hypothesis using embryonal neurulation. Some animals develop at lower or higher temperature.  How do the authors interpret this fact?

Author Response

REVIEWER 1

This manuscript proposes various concepts linking basic mathematical theories to some interesting topics (mostly, anatomical issues) in the nervous system.  The writing is good, although the topic is hypothetical. The mathematics includes geometry, topology, knot theory, braid groups, category theory, group theory, and even algebra and number theory. It also contains some cell biological, developmental biological, and physical phenomena.  

This manuscript has no experimental data, but the authors tried to formulate testable ideas. Those ideas were not experimentally proven in this manuscript, or were not subjected to meta-analysis based on previous data. They could be right or wrong. Some people may not feel that they are scientifically sound.  However, I enjoyed the authors’ ideas because many of them are inspiring. Most important, they are scientific.  Thus, I would hesitate to raise any concerns about each hypothesis that the authors proposed here. I have two major suggestions to help the readers understand the authors' proposals. 

First, I often felt that some ideas are presented without enough neuroscience background.  The authors often introduced theories, and then brought up real-world biology. Thus, it sounds strange at some places.  In general, I would suggest placing more standard neuroscience and biology (noticed ideas) before the authors introduce their “unnoticed” ideas. In this way, the authors’ ideas will be readily recognized as “unnoticed” ideas. 

Dear reviewer, thanks a lot your precious efforts. We tried to enlarge the preliminary, background descriptions concerning standard neuroscience and biology.  The changes we performed throughout the whole text are marked in green. 

Second, the authors tend to consider only humans.  However, other species also have similar structures.  I would suggest adding this aspect so that the authors' ideas will be more influential.  

Thanks.  We provided more animal background, in particular concerning the troublesome problem of visual depth.

Minor points

Line 46-51  Here, I wonder why the authors organized topics in this order.  Is there any scientific reason for this? Otherwise, this manuscript looks like a random collection of essays in which a series of interesting ideas are randomly arranged. 

Thanks for let us understanding the faults of our manuscript.  We arranged the whole discourse in a totally different manner, in order to make it more palatable and less random.  In particular, every chapter is now devoted to a different area of neuroscience, i.e., (macro-, meso- and micro-) neuroanatomy, nervous development, visual perception. 

Line 52 onwards, Geometry chapter

Monge’s theorem is interesting. There are many explanations about binocular and monocular visions (eg. saccade).  Most important, the authors consider visions of only human.  How about other species (eg, pigeon, owl)?  

We provided more neuroanatomical premises and more animal background. 

Line 95 onwards, Tunneling nanotubes chapter

This chapter focuses on a cell biological structure.  Thus, this part is different from other parts where the authors advocate mathematical theories. 

Standard neuroscience textbooks usually tell that neuron communicate via axons, dendrites, and synapses.  However, the authors do not provide this standard textbook view.  Many cell biologists would agree that we cannot dismiss tunneling nanotubes.  However, TNT should be presented together with the standard neuroscience view. 

This time, we clearly emphasized that the successful mathematical approach of neural networks might be unexpectedly extended also to the micro-scale of the neuronal membranes.

Line 110 nervus -> nervous

Thanks!

Line 146 onwards, Topology chapter (especially, line 186-)

The topological or topographical organization of neuronal connections is commonplace.  For example, in lower vertebrates, it is clear that topology plays important roles in visual perception. Here, I feel that the authors’ argument is not well balanced (line 175-).  

You are right.  Indeed, we decided to fully remove this controversial and too qualitative paragraph. 

Line 184, Several Authors -> Several authors

Line 194 chapter knot theory and braid groups 

The idea sounds interesting.  However, it is not so clear why knot theory and braid groups help understandneuroscience as the authors brought up (Line 246-).  The authors tried to explain their benefits. However, the benefits are not convincingly addressed because it looks the authors just presented a list of phenomena and theory. 

We provided novel paragraphs  to show how knots and braid groups might offer a scarcely explored approach to elucidate the arrangement of nervous fibers at the mesoscopic scales of the ascending pathways. 

Line 281- category theory and group theory

It is not clear that "Soft matter polymers and embryonal neurulation" is in "category theory and group theory". I like the testable hypothesis using embryonal neurulation. Some animals develop at lower or higher temperature.  How do the authors interpret this fact?

We fully changed the order and the content of the Sections.  We devoted to “Soft matter polymers and embryonal neurulation” a fully novel chapter entitled:  MATHEMATICS AND THE EMBRYONIC DEVELOPMENT OF THE NERVOUS SYSTEM.

Concerning the animals developing at different temperatures, we added a remark.  The fact that the embryonal neurulation occurs at different temperatures in different animals does not invalidate our theoretical account, since every species-specific embryogenic temperature might lead to anatomical modifications in the adult nervous system.    

Reviewer 2 Report

In general, the structure of this review manuscript is rather strange. It is not completely clear why the authors did chose to focus the attention to visual-perception models, and then jump to the patter-formation issues. Some more fluent transition between those two issues is required.

On the other hand, the discussion on visual-perception models is also somewhat biased. This research area is huge, including such aspects as modelling and control of visual perception, visual perception learning and models, visual perception and consciousness, static and motion-based visual illusions. The authors just pick few specific aspects of visual-perception models. 

It is understandable that the authors try to describe non-standard mathematical models and techniques used to to describe visual-perception models. However, some discussions should be addressed to the paradigmatic website "Optical Illusions & Visual Phenomena" https://michaelbach.de/ot/. A lot of nonstandard mathematics is available there. 

Another issue related to visual-perception models is related to mathematical aspects of dynamic visual cryptography. A typical recommended reference in this area is: Visual integration of vibrating images in time. Optical Engineering. 2018, vol.57(9), article no.093107.

And, as mentioned previously, the transition from visual perception to the formation of nervous tissue should be re-arranged in a more fluent manner. 

A major revision is recommended. 

Author Response

REVIEWER 2

In general, the structure of this review manuscript is rather strange. It is not completely clear why the authors did chose to focus the attention to visual-perception models, and then jump to the patter-formation issues. Some more fluent transition between those two issues is required.

Dear reviewer, thanks a lot.  Your wise comment let us realize the flaws of our manuscript.  We arranged the whole discussion in a totally different way, in order to make it less confused and more fluid.  In particular, every chapter is now devoted to a different area of neuroscience, i.e., (macro-, meso- and micro-) neuroanatomy, nervous development, visual perception.   All the changes throughout the whole text that are marked in green.

On the other hand, the discussion on visual-perception models is also somewhat biased. This research area is huge, including such aspects as modelling and control of visual perception, visual perception learning and models, visual perception and consciousness, static and motion-based visual illusions. The authors just pick few specific aspects of visual-perception models. 

It is understandable that the authors try to describe non-standard mathematical models and techniques used to to describe visual-perception models. However, some discussions should be addressed to the paradigmatic website "Optical Illusions & Visual Phenomena" https://michaelbach.de/ot/. A lot of nonstandard mathematics is available there. 

Another issue related to visual-perception models is related to mathematical aspects of dynamic visual cryptography. A typical recommended reference in this area is: Visual integration of vibrating images in time. Optical Engineering. 2018, vol.57(9), article no.093107.

And, as mentioned previously, the transition from visual perception to the formation of nervous tissue should be re-arranged in a more fluent manner. 

We introduced the important references and provided an effort to enlarge our preliminary discussion concerning visual depth, inserting more neuroanatomical background and discussing the state of the art.

Round 2

Reviewer 2 Report

The authors did make a good revision of the original manuscript. It can be recommended for publication in the present form. 

Author Response

Dear Reviewer,

thanks a lot!